# A Near Real-Time and Free Tool for the Preliminary Mapping of Active Lava Flows during Volcanic Crises: The Case of Hotspot Subaerial Eruptions

Francisco Javier Vasconez [1,*] , Juan Camilo Anzieta [2,3] , Anais Vásconez Müller [1] , Benjamin Bernard [1] and Patricio Ramón [1]

1 Instituto Geofísico, Escuela Politécnica Nacional, Quito 170525, Ecuador; avasconez@igepn.edu.ec (A.V.M.); bbernard@igepn.edu.ec (B.B.); pramon@igepn.edu.ec (P.R.)
2 Department of Earth Sciences, Simon Fraser University, Surrey, BC V5A 1S6, Canada; janzieta@sfu.ca
3 Escuela de Ciencias Físicas y Matemática, Pontificia Universidad Católica del Ecuador, Quito 170525, Ecuador
* Correspondence: fjvasconez@igepn.edu.ec

**Abstract:** Monitoring the evolution of lava flows is a challenging task for volcano observatories, especially in remote volcanic areas. Here we present a near real-time (every 12 h) and free tool for producing interactive thermal maps of the advance of lava flows over time by taking advantage of the free thermal data provided by FIRMS and the open-source R software. To achieve this, we applied two filters on the FIRMS datasets, one on the satellite layout (track) and another on the fire radiative power (FRP). To determine the latter, we carried out a detailed statistical analysis of the FRP values of nine hotspot subaerial eruptions that included Cumbre Vieja-2021 (Spain), Fagradalsfjall-2021 (Iceland), LERZ Kilauea-2018 (USA), and six eruptions on the Galápagos Archipelago (Ecuador). We found that an FRP filter of $35 \pm 17$ MW/pixel worked well at the onset and during the first weeks of an eruption. Afterward, once the cumulative statistical parameters had stabilized, a filter that better fit the investigated case could be obtained by running our statistical code. Using the suggested filters, the thermal maps resulting from our mapping code have an accuracy higher than 75% on average when compared with the official lava flow maps of each eruption and an offset of only 3% regarding the maximum lava flow extension. Therefore, our easy-to-use codes constitute an additional, novel, and simple tool for rapid preliminary mapping of lava fields during crises, especially when regular overflights and/or unoccupied aerial vehicle campaigns are out of budget.

**Keywords:** lava flow; mapping; FIRMS; VIIRS; volcanic crises; thermal sensing; effusive eruption

## 1. Introduction

According to Lowenstern et al. [1], during volcanic crises, observatories have to acquire, process, analyze, and interpret data in a timely manner to effectively communicate the current situation and potential future hazard scenarios to decision makers and the public. In the best-case scenario, volcano observatories analyze data from both ground-based geophysical stations and satellite-derived information. Geophysical instruments can include seismic, acoustic, outgassing, and ground deformation ones, among others, that usually transfer information in real-time. On the other hand, optical and/or radar satellite sensors transmit data in near real-time or periodically, and their most used associated products allow the detection of ash dispersion, thermal anomalies, $SO_2$ emissions, and ground deformation, among others, which are used by volcano observatories worldwide to aid the assessment of volcanic activity.

Ground-based information can be scarce for remote volcanic areas because of transmission and maintenance issues, especially in low-income countries [2]. In contrast, the newly available satellite imagery (free or for academic use) in addition to the development of various online platforms such as: Volcanic Cloud Monitoring [3,4], FIRMS [5–8],

MIROVA [9,10], MODVOLC [11], WORLDVIEW (https://worldview.earthdata.nasa.gov), MOUNTS [12], the Global Sulfur Dioxide Monitoring [13,14], Planet [15], and the International Disaster Charter (https://disasterscharter.org/web/guest/home), have significantly improved the surveillance by satellite, and therefore enhanced the short-term hazard assessment, especially in low-income countries and in remote volcanic areas, e.g., [16–19].

Thermal anomaly detection has been a very useful tool during recent effusive volcanic crises around the world since it allows us to detect signs of unrest [20,21], follow the evolution of the eruption and calculate lava effusion rates, cumulative volume, and the extension of lava flows [16–19,22–30]. In this context, we propose a near real-time (twice per day) and free-to-use tool for mapping active lava flows of subaerial hotspot-related eruptions through time. Our scripts take advantage of the freely-available thermal FIRMS data (https://firms.modaps.eosdis.nasa.gov) and the open-source R software (https://www.r-project.org/) to generate interactive preliminary maps of the areas covered by fresh lava fields. We compared our results with satellite images and the official lava flow maps of six Galápagos eruptions in addition to the most recent eruptions of Cumbre Vieja-2021 (Spain), Fagradalsfjall-2021 (Iceland), and Kilauea-2018 (USA) to calibrate and validate the tool. Our codes provide an additional tool to constrain the area most likely covered by fresh lava in near real-time, which is especially useful in the context of volcanic crises, specifically during effusive and remote eruptions, and is intended for use by volcano observatories with financial limitations.

## 2. Materials and Methods

The Visible Infrared Imaging Radiometer Suite (VIIRS) instrument on board the joint NASA/NOAA satellites Suomi National Polar-orbiting Partnership (S-NPP), with data since 2012, and the NOAA-20/JPSS-1, with temporal coverage since 2020 [31,32], provides continued fire radiative power (FRP) data twice per day with a pixel size of 370 m [7,33,34] by using the thermal band 0.412–11.5 μm. FRP is the emitted radiant power in Megawatt/pixel (MW/pixel) released during combustion [35] and has been utilized to estimate biomass burning emissions, combustion rates, burned areas, and wildfire size, among others [35–45]. The VIIRS data is freely processed and distributed by NASA via the webpage Fire Information from Resource Management System, FIRMS (https://firms.modaps.eosdis.nasa.gov, accessed on 14 March 2022) within three hours of the satellite observation, as a level-2 satellite-based product. Data includes acquisition date, location, and FRP, among other parameters related to satellite data acquisition, such as track and scan GSD (ground sample distances). These data can be freely downloaded from the archive, which covers the world, country, custom regions, and protected areas. Fire sources include MODIS, VIIRS S-NPP, and VIIRS NOAA-20 sensors, and can be downloaded in shapefile (.shp), comma-separated text (.csv) and JSON (.json) formats providing an email (Supplementary Material S1A).

Here we used data from VIIRS S-NPP and VIIRS NOAA-20 in ".csv" format to be read, filtered, analyzed, and plotted in the R/RStudio software (Supplementary Material S1B). We developed two dedicated scripts, one to analyze the FRP values statistically and another to plot the thermal anomalies on top of a basemap (e.g., Esri.WorldImagery). Our codes produce one composite figure with histograms and cumulative statistical time-series of the FRP evolution for the period of acquisition, one dynamic thermal map of the lava flow progression over time, and one figure of the maximum length of the lava (in kilometers) in reference to the uppermost vent location (if available) or another referential point. Additionally, six tables are saved that include two files with the statistical summary of the FRP FIRMS' data, two files with the filtered data by satellite, and two files with the maximum lava distance over time for each satellite (i.e., S-NPP and NOAA-20) for further analysis. To calibrate the codes, we explored six eruptions that occurred during the last decade in the Galápagos Archipelago, i.e., (2) Wolf-2022 and -2015, (1) Sierra Negra-2018, and (3) Fernandina-2020, -2018, and -2017, in addition to Cumbre Vieja (Spain) and Fagradalsfjall (Iceland) eruptions of 2021 and the 2018 Lower East Rift Zone (LERZ)—

Kilauea eruption (USA). Figure 1 shows a simplified workflow of the input data and methodology. A detailed description of the steps to be taken for the implementation of this tool is given below and in Supplementary Material S1.

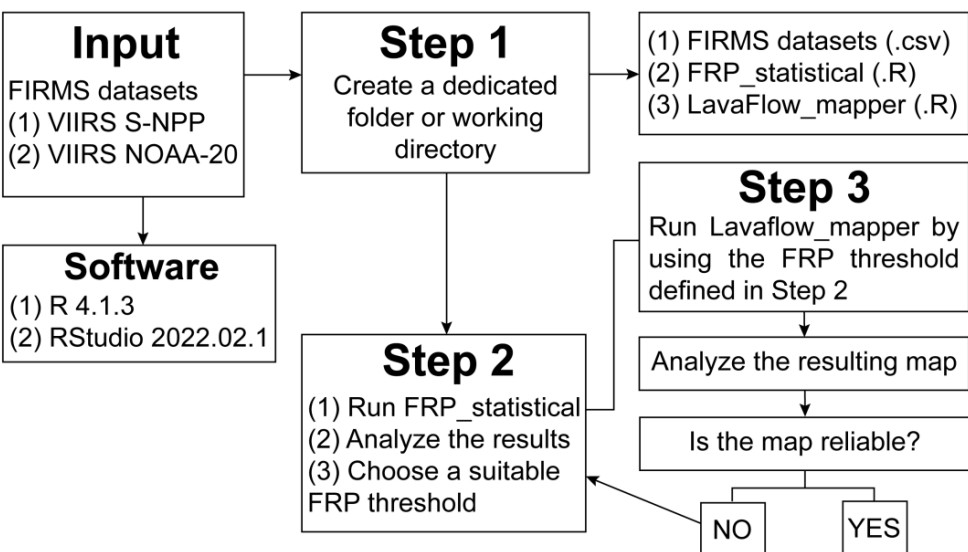

**Figure 1.** Simplified workflow of our tool for mapping lava flows in near real-time. In this context, reliability is defined by comparing the results of our tool with satellite or UAV images.

*2.1. Input Parameters and Workspace*

Before compiling the codes, it is necessary to download the FIRMS S-NPP and NOAA-20 datasets of the eruption being investigated, i.e., the thermal anomalies contained in the area of interest, between the onset and end of the eruption for past eruptions or from the onset to the processing day for ongoing eruptions (Supplementary Material S1A). Moreover, it is useful to know the approximate location of the uppermost vent in decimal WGS84 coordinates (i.e., longitude and latitude) or to have a reference point that will be utilized to estimate the maximum linear extent of the lava (i.e., the maximum radius with respect to the given point). The two files provided by FIRMS in ".csv" format and our two R-scripts (Supplementary Material S1C or https://vhub.org/tools/lavaflowmapper/wiki, last accessed on 13 June 2022) should be located in a single dedicated folder that makes up the workspace (Figure 1). Other R-users can refer to dedicated folders for data storage and are free to alter the reading/saving part of the code as needed. Note: We recommend not to use a space in the folder name to avoid errors; instead, the underscore is a better option (e.g., "folder_name", "Kilauea_2018", etc.).

*2.2. First Set up in R/RStudio*

For the first run, it is required to install R-4.1.3 (Supplementary Material S1B) with the following packages: "tictoc", "leaflet", "leaflegend", "sp", "viridisLite", "viridis", "mapview", "geosphere", and "webshot". For simplicity, we strongly suggest installing the interface RStudio 2022.02.1 Build 461 or later, where, after opening our scripts, the software will automatically identify the need for these packages and will ask the user to install them (see Supplementary Material S1B). If an error occurs, be sure that the packages were correctly installed or check compatibility between versions of R, RStudio, additional requirements (e.g., XQuartz for Mac, etc.), and the packages being installed.

*2.3. Run FRP_Statistical*

The FRP_statistical script provides histograms and statistical cumulative time-series of the fire radiative power of each database (i.e., S-NPP and NOAA-20). The histograms also display a summary of statistical parameters that include mean, minimum, 5th percentile, 1st quartile, median, 3rd quartile, 95th percentile, and maximum values, in addition to a

black dashed line to highlight the mean and a blue dotted line to indicate the 3rd quartile (Figure 2a). Moreover, statistical time series on the cumulative FRP are plotted at the bottom of the figure to visualize the variations of the mean, 1st quartile, median, and 3rd quartile every 12 h throughout the eruption since the continuous data acquisition may affect these values as the eruption continues (Figure 2b). The resulting composite figure allows us to understand the fire radiative power values better and thus to define a suitable minimum FRP threshold to be later used in the LavaFlow_mapper code as a filter. In hindsight, this is done via trial and error by comparing the thermal maps resulting from the application of each filter with satellite imagery and the official lava flow maps of the eruptions being investigated, published by the respective institutions in charge in each country (see Section 3).

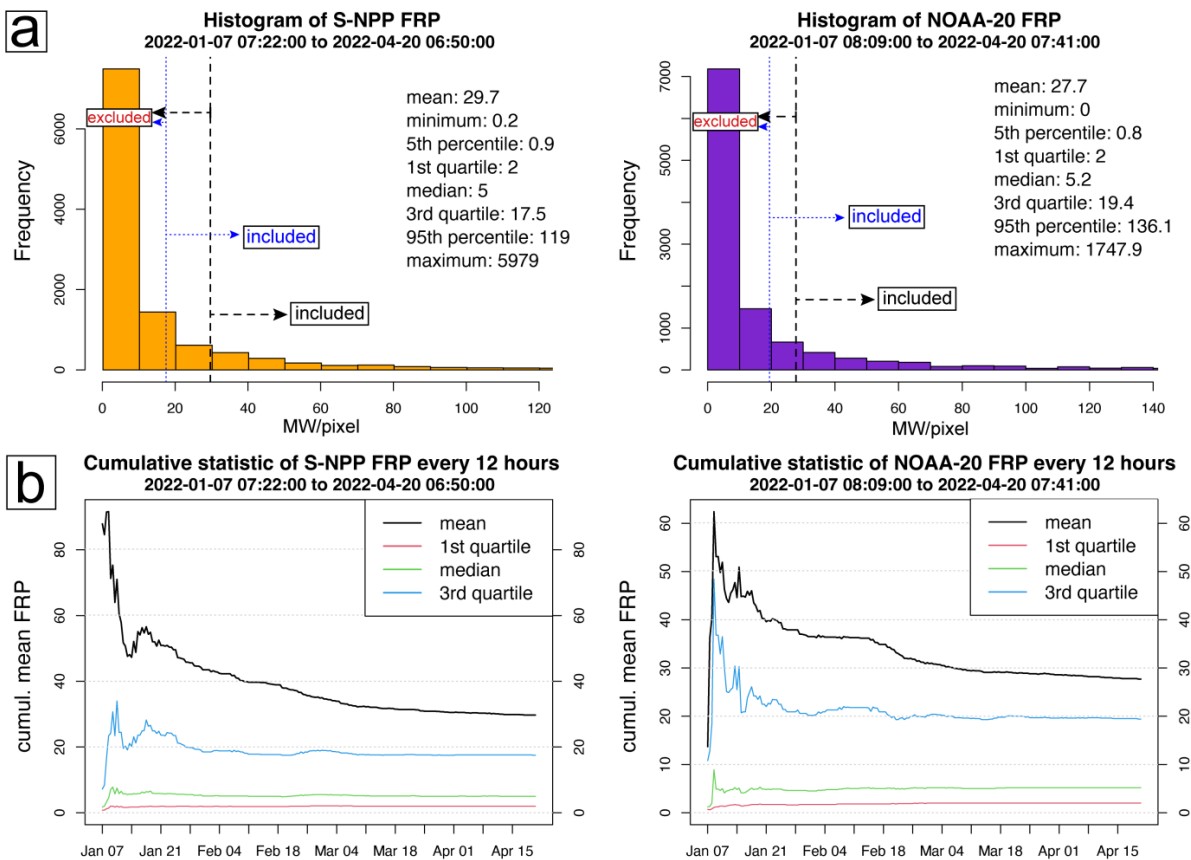

**Figure 2.** Resulting composite figure of the statistical analysis of the fire radiative power (FRP) for the Wolf 2022 eruption (a compilation of the resulting statistical analysis for all case studies is found in Supplementary Material S2A). (**a**) Two histograms per satellite that summarize the statistics of the entire dataset. Black dashed line indicates the mean; blue dotted line is the 3rd quartile. When used as filters, values with FRP below the chosen statistical value are excluded, and values above it are included for mapping in the second code. (**b**) Temporal variations of various cumulative FPR statistical parameters every 12 h (i.e., from the onset to the processing day).

The FRP statistical script can be utilized as many times as the user needs since the FIRMS database is continuously fed with new data, i.e., the cumulative statistical parameters change over time (Figure 2b). As a result, the "Plot" tab will display the composite figure, which can be exported as a .png image or pdf. In addition, two csv files with the cumulative statistical analysis of the data retrieved every 12 h by each satellite will be automatically saved in the workspace. Note that the working directory is automatically set up by our codes and corresponds to the path of the folder on the user computer in which the scripts are located (see Section 2.1)—a default setting that can be changed by the user if

needed. We strongly recommend having a dedicated folder for each case being investigated to avoid file reading conflicts (e.g., Wolf_2022, Wolf_2015, CumbreVieja_2021, etc.).

### 2.4. Run LavaFlow_Mapper

In general, filter application implies a significant reduction of data under given conditions. The LavaFlow_mapper code applies two filters on the original FIRMS data to generate a dynamic temporal map, a figure, and four resulting tables. The first filter is applied on the track GSD (i.e., satellite layout), and the second one is on the fire radiative power (i.e., radiant energy). For the former, Wang et al. [46] performed a detailed analysis of the VIIRS S-NPP and NOAA-20 data to improve the geolocation of the thermal anomalies. They concluded that a filter on the track parameter lesser or equal to 0.5 is suitable to discard wrong geolocation of thermal anomalies due to the satellite's view zenith angle. The second filter is applied to the FRP data and consists of applying the threshold previously defined with the FRP_statistical script (see Section 2.3), which depends on the characteristics of the eruption being investigated. Moreover, a reference point in decimal WGS84 coordinates that indicates the lava origin (i.e., vent location) must be set to estimate the maximum linear extent of the lava flows per satellite and through time. If the vent location is not available or the estimation of the lava extent is not needed, the default value should be kept for code consistency. However, the user should be aware that this result is no longer relevant. Finally, the LavaFlow_mapper will display an interactive map in the "Viewer" and a figure in the "Plots" tabs, respectively. In both cases, the plots can be exported as .png figures or .pdf files. Additionally, four csv files are produced: (1) filter_VIIRS_S-NPP.csv, (2) filter_VIIRS_NOAA-20.csv, (3) distance_SNPP.csv, and (4) distance_NOAA.csv, which can be used for further analysis if needed.

## 3. Case Studies

In this section, we summarize the six most recent and remote eruptions of the Galápagos Archipelago from 2015 to 2022 and the most recent and major eruptions of Cumbre Vieja (Spain) and Fagradalsfjall (Iceland) in 2021, and the Lower East Rift Zone (LERZ)—Kilauea eruption (USA) in 2018. These nine hotspot eruptions were chosen as case studies to calibrate and validate our tool.

### 3.1. Wolf Eruptions, Galápagos—Ecuador

Wolf volcano (0.042°N; 91.335°W; 1705 m a.s.l.) is located on the northern tip of Isabela Island (Figure 3a). Since 1797 AD, it has erupted ~13 times [27,47] and is therefore considered one of the most active volcanoes of the Galápagos Archipelago [27,48]. Due to its low associated risk, its remote location and the lack of road access, this volcano does not have its own ground-based monitoring network. Nonetheless, the FER1 broadband seismic station located on Fernandina Island (33 km southwest of the volcano, Figure 3a) is currently utilized to monitor Wolf's activity [49] by using matched filtering [18,50].

#### 3.1.1. Wolf—2015 Eruption

The eruption began on 25 May 2015 through a ~3 km circumferential fissure on the southeastern and eastern flanks of the caldera rim (Figure 3a). The eruption had two main phases; the first extruded lava through the circumferential fissure from 25 May to 16 June, while the second expelled lava within the caldera between 13 and 30 June [27]. In 36 days, lava covered an estimated area of ~18.5 km$^2$ at the SE and E flanks and ~5.9 km$^2$ inside the caldera (Table 1), which implies a total area of 24 ± 1.5 km$^2$ and a bulk volume of 116 ± 45 million m$^3$ [27].

#### 3.1.2. Wolf—2022 Eruption

On 6 January 2022, after seven years of quiescence, Wolf began a new eruptive period through a 7–8 km long radial fissure on the southeastern flank, on the border of the caldera (azimuth ~130°, Figure 3a). The eruption, still ongoing in April 2022 (103 days until

20 April), is characterized by the continuous emission of lava flows. At the onset of the eruption, two ash-poor eruptive columns of 5.5 and 3.3 km a.s.l. were reported by the Washington VAAC. These columns were gas-rich and expelled ~49.7 ± 11 kilotons of $SO_2$ on average, as reported by MOUNTS and the Global Sulphur Dioxide Monitoring (NASA) via TROPOMI, OMPS, and OMI satellite sensors. Since the start of the eruption, several pulses of activity have been recognized, and the most intense ones occurred during the first two weeks, when lava reached ~15 km from the upper vent. Until 20 April, lava had covered ~30 km² (Table 1 and Figure 3a).

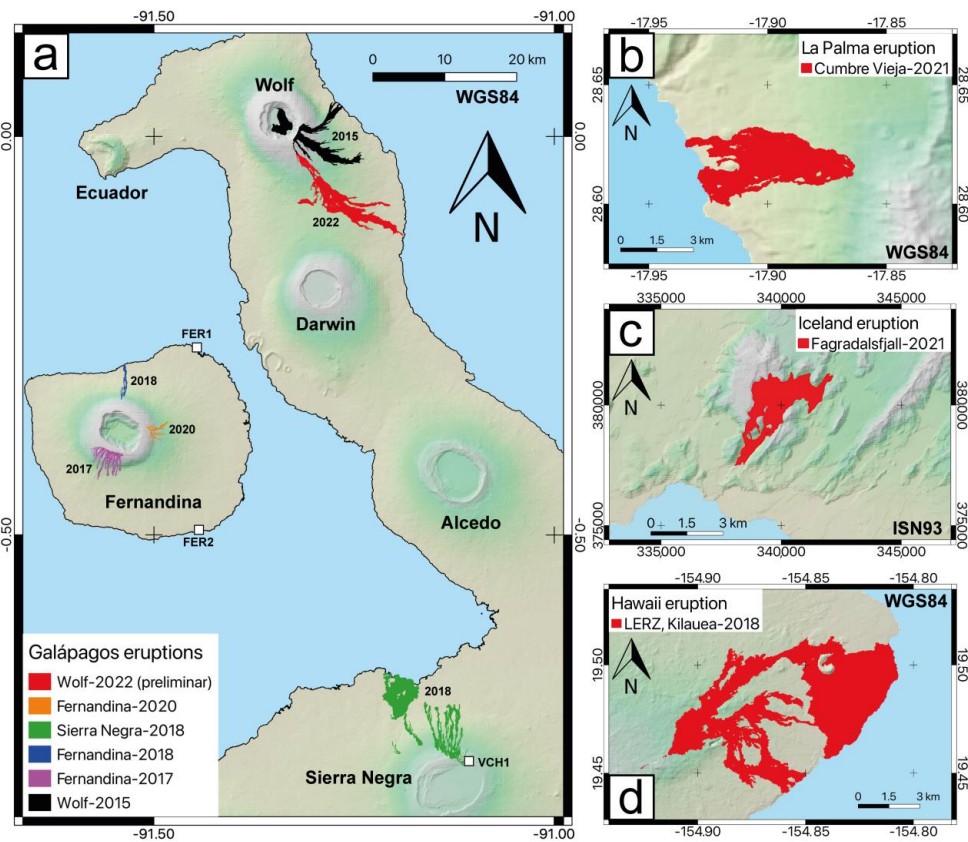

**Figure 3.** (**a**) Galápagos eruptions between 2015 and 2022. White squares highlight the location of the geophysical monitoring stations, (**b**) Cumbre Vieja eruption in 2021, (**c**) Fagradalsfjall eruption in 2021, and (**d**) LERZ-Kilauea eruption in 2018. Polygons depict the area covered by the lava according to the corresponding official institutions (see Table 1 for references).

### 3.2. Sierra Negra—2018 Eruption, Galápagos—Ecuador

The Sierra Negra volcano (0.782°S, 91.139°W, 1139 m a.s.l.) is located in the southern region of Isabela Island (Figure 3a). On average, it has one eruptive period every 12 years, and most of its recent activity has impacted the uninhabited northern flank and the caldera floor [51]. One multiparametric station located on Volcán Chico (VCH1) and a network of high-precision GPS (cGPS) permanently monitor the volcano [52]. After 13 years of quiescence and one year of persistent unrest, the Sierra Negra volcano began a new eruptive period on 26 June 2018, which lasted 58 days (Table 1). According to Vasconez et al. [52], the eruption was characterized by the emission of lava flows toward the north and northwestern flanks of the volcano via five eruptive fissures that were active in two main phases. The first one, which lasted less than a day, was the most intense one and included eruptions from the five eruptive fissures, while the second phase was the longest one and centered on the lowermost fissure [52]. At the end of the eruption, lava covered ~30.6 km² and expelled an estimated bulk volume of 189 ± 94 million m³ [52] (Table 1).

**Table 1.** Summary of the nine case studies related to subaerial hotspot eruptions.

| Eruption (Country) | First Day | Last Day | Duration (Days) | Main Phenomena | Area (km²) | Volume (Million m³) | Reference |
|---|---|---|---|---|---|---|---|
| Wolf-2022 (Ecuador) | 6 January | 20 April [1] | >100 | Lava flows | ~30 | - | This study |
| Wolf-2015 (Ecuador) [2] | 25 May | 30 June | 36 | Lava flows + gas plumes | 24 ± 1.52 | 116 ± 45 | [27] |
| Sierra Negra-2018 (Ecuador) [2] | 26 June | 23 August | 58 | Lava flows | 30.6 | 189 ± 94 | [52] |
| Fernandina-2020 (Ecuador) | 12 January | 13 January | 0.4 | Lava flows | 1.63 | - | This study |
| Fernandina-2018 (Ecuador) [2] | 16 June | 18 June | 2 | Lava flows | 1.58 | 7.9 ± 4 | [52] |
| Fernandina-2017 (Ecuador) | 4 September | 7 September | 2.5 | Lava flows + wildfires | 6.5 | 13 ± 6.5 | [52] |
| Cumbre Vieja-2021 (Spain) [2] | 19 September | 14 December | 85.3 | Lava flows + tephra plumes | 12.4 | >200 | IGN & EMS; [53] |
| Fagradalsfjall-2021 (Iceland) | 19 March | 18 September | 183 | Lava flows | 4.8 | 150 ± 3 | [54,55] |
| LERZ, Kilauea-2018 (USA) [2] | 18 May | 4 August | 78 | Lava flows + gas plumes | >35.6 | 1179 ± 97 | [56–58] |

[1] We took 20 April as the preliminary cut-off date, while the exact date of the end of the eruption was still unknown. [2] Lava reached the sea.

### 3.3. Fernandina Eruptions, Galápagos—Ecuador

Fernandina volcano (0.353°S, 91.525°W, 1481 m a.s.l.) is the westernmost volcano of the Galápagos Archipelago (Figure 3a). It has had at least 29 historical eruptions since the 1800s, making it one of the most active volcanoes of the Galápagos Islands [52,59]. Despite having no populated areas on its lower flanks, Fernandina is one of the most visited islands by tourists and scientists due to its rich flora, fauna, and geology [60]. In 2014, the IG-EPN installed two permanent broadband seismic stations with satellite transmission to surveil the volcano in real-time, FER1 and FER2 (Figure 3a).

#### 3.3.1. Fernandina—2017 Eruption

After 8 years of quiescence, on 4 September 2017, a short seismic unrest period was registered by the IG-EPN network. At 18h25 UTC, a tremor dominated the seismic record indicating the onset of the eruption [52]. The lava was expelled from a 3 km-long circumferential fissure on the southwestern flank (Figure 3a) for two and a half days. They covered 6.5 km² of land with an approximated bulk volume of 13 ± 6.5 million m³ [52] (Table 1). The most long-lasting hazard associated with this eruption was the wildfires triggered by the interaction of the hot lava flows and the flora [52]. Wildfires were active until a month after the eruption ended and consumed more than 17 km² of native flora at the western flank of the volcano [52].

#### 3.3.2. Fernandina—2018 Eruption

On 16 June 2018, after 9 months of repose, Fernandina erupted throughout a radial fissure on the upper northern flank (Figure 3a). Before the eruption, the IG-EPN seismic network detected short seismic unrest characterized by nine earthquakes larger than M2.5 [52]. The eruption lasted 2 days, in which lava covered 1.58 km² and reached the sea, with a total estimated volume of 7.9 ± 4 million m³ [52] (Table 1).

#### 3.3.3. Fernandina—2020 Eruption

After 19 months of quiescence, Fernandina commenced a new eruptive period on 12 January 2020, after presenting ground deformation and seismic unrest characterized by six earthquakes up to M4.3 [49]. The eruption took place through a circumferential fissure

on the eastern caldera rim (Figure 3a) and lasted ~9 h. Based on a Sentinel-2 image from 2 March 2020, we estimated that the lava covered approximately 1.63 km$^2$ (Table 1).

### 3.4. Cumbre Vieja—2021 Eruption, Canary Islands—Spain

The Cumbre Vieja volcano (28.57°N, 17.83°W, 2426 m a.s.l.) is located on La Palma, Canary Islands, Spain (Figure 3b). Since historical times, i.e., the 15th century, it has erupted eight times, implying an average recurrence period of one eruption every 67 years [61]. Its most recent activity has impacted the populated areas to the west through mild explosive activity and lava flows [61,62]. In La Palma, the Instituto Geográfico Nacional (IGN) monitoring network, which includes, among others, seismic and GPS stations, has detected multiple unrest signals since 2017 [62]. On 11 September 2021, a seismic swarm characterized by a sustained increase in the number and magnitude of earthquakes, in addition to more than 20 cm of uplift, was detected [62,63]. On 19 September 2021, after 50 years of quiescence, the Cumbre Vieja volcano commenced a new eruptive period. The eruption began through two fissures and multiple vents that produced lava fountains, tephra plumes and lava flows that traveled toward the west, destroying hundreds of properties [64]. The eruption lasted three months, and the lava covered 12.4 km$^2$ [65] (Table 1). Periodic maps of the progression of the lava flows were elaborated and published by IGN in partnership with Copernicus Emergency Management Service (EMS) by using optical satellite imagery. Additionally, a bulk volume of more than 200 million m$^3$ was estimated for the entire eruption [53].

### 3.5. Fagradalsfjall-2021 Eruption, Iceland

In Iceland, there is, on average, one eruption every 3–5 years [66]. However, on the western tip of the island, at the Reykjanes Peninsula, no eruptions have occurred in the last ~800 years [67]. The peninsula comprises five volcanic systems: the Reykjanes, Svartsengi, Krýsuvík, Fagradalsfjall, and Brennisteinsfjöll. The Fagradalsfjall system appeared to be the least active of the five, and its last eruption occurred over 6000 years ago [67]. Nevertheless, after more than a year of volcano-tectonic unrest, which was timely identified, monitored, and reported by the Icelandic Meteorological Office (IMO) and colleague institutions, on 19 March 2021, an eruption occurred in the Fagradalsfjall area (Figure 3c). The Fagradalsfjall eruption (63.917°N, 22.067°W, 360 m a.s.l.) was characterized by the emission of lava flows via six fissures during four eruptive phases that ended on 18 September 2021 [54,68]. In total, lava covered 4.8 km$^2$ (Table 1) with an estimated bulk volume of $150 \pm 3$ million m$^3$ [55]. Near real-time stereo-photogrammetry by combining satellite and airborne stereo images allowed us to map the progression of lava flows over time and monitor key eruption parameters such as effusion rates and cumulative volume [55].

### 3.6. Lower East Rift Zone (LERZ), Kilauea—2018 Eruption, Hawai'i—USA

The Kilauea volcano (19.421°N, 155.287°W, 1222 m a.s.l.) is located on the southeastern end of Hilo Island, Hawai'i, USA (Figure 3d). Kilauea has been erupting from two vents: a lava lake within Halema'uma'u summit crater since 2008 [69] and the Pu'u 'Ō'ō cone and nearby vents in the East Rift Zone since 1983 [70,71]. Because of the easy accessibility and dense monitoring network, Kilauea has extensive long-term databases, which have provided new insights into how calderas and rift systems interact [57,72]. In 2018, Kilauea volcano experienced its largest Lower East Rift Zone (LERZ) eruption and caldera collapse in the last 200 years [57]. The LERZ eruption extruded a total volume of $1179 \pm 97$ million m$^3$ of lava [56] through 24 eruptive fissures that covered more than 35.6 km$^2$ [57,58] (Table 1). The eruption lasted ~78 days; periodically, maps of the advance of the lava flows and Digital Elevation Models (DEMs) were created with the help of crewed fixed-wing, helicopter, and unoccupied aerial vehicle overflights [56]. The long-term monitoring, as well as the timely response and warning of the eruption, were key to avoiding causalities in the cascading 2018 Kilauea eruption [72].

## 4. Results

### 4.1. FRP_Statistical

Two approaches were contemplated for the FRP statistical analysis. In the first one, we ran the code over the entire dataset and obtained the summary of the statistical parameters, which is shown as a table within the histogram (Figure 2a and Table 2). For the second approach, we considered the periodical addition of new data and developed a temporal analysis of the cumulative FRP statistical variations in a half-day manner since satellites pass twice per day (Figure 2b).

**Table 2.** Summary of the fire radiative power (MW/pixel) statistical parameters: mean, minimum, 5th percentile, 1st quartile, median, 3rd quartile, 95th percentile, and maximum for each case study considering the entire dataset.

| Eruption-Year | Satellite | Duration | Mean | Min. | 5th Per. | 1st Quar. | Median | 3rd Quar. | 95th Per. | Max. | # Anomalies |
|---|---|---|---|---|---|---|---|---|---|---|---|
| Wolf-2022 [1] | S-NPP | 103 | 30 | 0 | 1 | 2 | 5 | 18 | 119 | 5979 | 11,498 |
| | NOAA-20 | | 28 | 0 | 1 | 2 | 5 | 19 | 136 | 1748 | 11,426 |
| Wolf-2015 | S-NPP | 36 | 56 | 0 | 1 | 2 | 7 | 27 | 337 | 1306 | 5269 |
| Sierra Negra-2018 | S-NPP | 58 | 29 | 0 | 1 | 2 | 5 | 17 | 153 | 1024 | 8628 |
| Cumbre Vieja-2021 | S-NPP | 85.3 | 34 | 0 | 1 | 2 | 6 | 25 | 185 | 719 | 15,060 |
| | NOAA-20 | | 31 | 0 | 1 | 2 | 5 | 18 | 165 | 1590 | 15,225 |
| Fagradalsfjall-2021 | S-NPP | 183 | 42 | 0 | 1 | 3 | 9 | 43 | 195 | 914 | 9918 |
| | NOAA-20 | | 41 | 0 | 1 | 2 | 8 | 36 | 198 | 1451 | 9658 |
| LERZ, Kilauea-2018 | S-NPP | 78 | 64 | 0 | 1 | 3 | 9 | 39 | 379 | 2089 | 34,628 |
| **Average long-lasting** | | 91 ± 51 | 39 ± 13 | 0 ± 0 | 1 ± 0 | 2 ± 0 | 7 ± 2 | 27 ± 10 | 207 ± 90 | | $10^3$–$10^4$ |
| Fernandina-2020 | S-NPP | 0.4 | 44 | 2 | 10 | 21 | 33 | 53 | 126 | 126 | 47 |
| | NOAA-20 | | 16 | 3 | 4 | 11 | 13 | 23 | 26 | 26 | 26 |
| Fernandina-2018 | S-NPP | 2 | 74 | 1 | 1 | 2 | 5 | 26 | 475 | 2689 | 392 |
| Fernandina-2017 | S-NPP | 2.5 | 116 | 1 | 2 | 6 | 14 | 44 | 346 | 4334 | 528 |
| **Average short-lived** | | 2 ± 1 | 63 ± 43 | 2 ± 1 | 4 ± 4 | 10 ± 8 | 16 ± 12 | 37 ± 14 | 243 ± 204 | | |
| **TOTAL average** | | 63 ± 55 | 44 ± 26 | 0 ± 1 | 2 ± 2 | 4 ± 5 | 9 ± 7 | 28 ± 12 | 209 ± 122 | | $10^1$–$10^2$ |

[1] We took 20 April as the preliminary cut-off date. The exact date of the end of the eruption was still unknown. # stands for number.

For the first case, we split our case studies into two groups: long-lasting and short-lived eruptions, where long-lasting ones lasted more than 30 days, and short-lived ones lasted from a few hours to 2.5 days (Table 2). For the short-lived eruptions, most of the statistical parameters (excluding the minimum) vary strongly between eruptions, and the number of thermal anomalies is in the order from tens to hundreds (Table 2). On the other hand, for long-lasting eruptions, minimum, 5th percentile, 1st quartile, and median are almost constant or vary slightly, while mean, 3rd quartile, 95th percentile, and maximum show significant differences between eruptions (Table 2). The total number of thermal anomalies is in the order of thousands and increases with the duration of the eruption up to tens of thousands (Table 2). Since the aim of the FRP statistical analysis is to define a minimum FRP threshold to filter the FIRMS data, the various statistical parameters are considered as possible thresholds. The minimum, 5th percentile, 1st quartile, and median generally fall in low FRP values, while the 95th percentile and the maximum fall in high ones (Figure 2a, Table 2, and Supplementary Material S2A). When looking at the exemplary histograms in Figure 2a, it can be seen that, if the lower values were taken as the minimum threshold, most thermal anomalies would be considered, while using the higher ones would significantly reduce the number of analyzed hot spots. Consequently, using low thresholds will allow more noise in the dataset to be used for the creation of the thermal maps, leading to unrealistically overestimated lava flow extensions, while increasing the threshold will allow cleaner data for the analysis with the risk of producing underestimated maps. From the exploration of the many possibilities, it seems that the mean and the 3rd quartile are the most suitable thresholds, as they fall in between the extremes of the histograms (Figure 2a), and their application implies the consideration of a moderate number of thermal anomalies for plotting the thermal map that also includes the highest FRP values, which certainty correspond to the hottest active lava.

In the second approach, we explored how these statistical parameters vary over time, especially the mean and the 3rd quartile, as they appeared to be the best fit for a possible FRP threshold. We focused on the six long-lasting eruptions, including the ongoing Wolf-2022 eruption, and calculated the cumulative statistical variables in a half-day cumulative manner to obtain the wanted parameters. In each case study, we observed that the cumulative mean and the 3rd quartile fluctuate strongly during the first days of the eruption and then tend to stabilize over time while the minimum, 1st quartile, median, and maximum are almost constant throughout the entire studied period (Supplementary Material S2A). As an example, Figure 4 shows how the mean (Figure 4a) and the 3rd quartile (Figure 4b) vary throughout the eruptions. Overall, each eruption produced a different curve, where increasing trends point out eruptive pulses, while stable and decreasing trends mark breaks or the end of the eruptions (Figure 4). Since each case has its own characteristics, i.e., its own cumulative statistical and physical parameters, the chosen FRP filter is different for each case study. Nevertheless, it varies within a narrow range ±20 MW/pixel (Figure 4), considering that raw FRP values range from 0 to >1000 MW/pixel (Table 2 and Supplementary Material S3A).

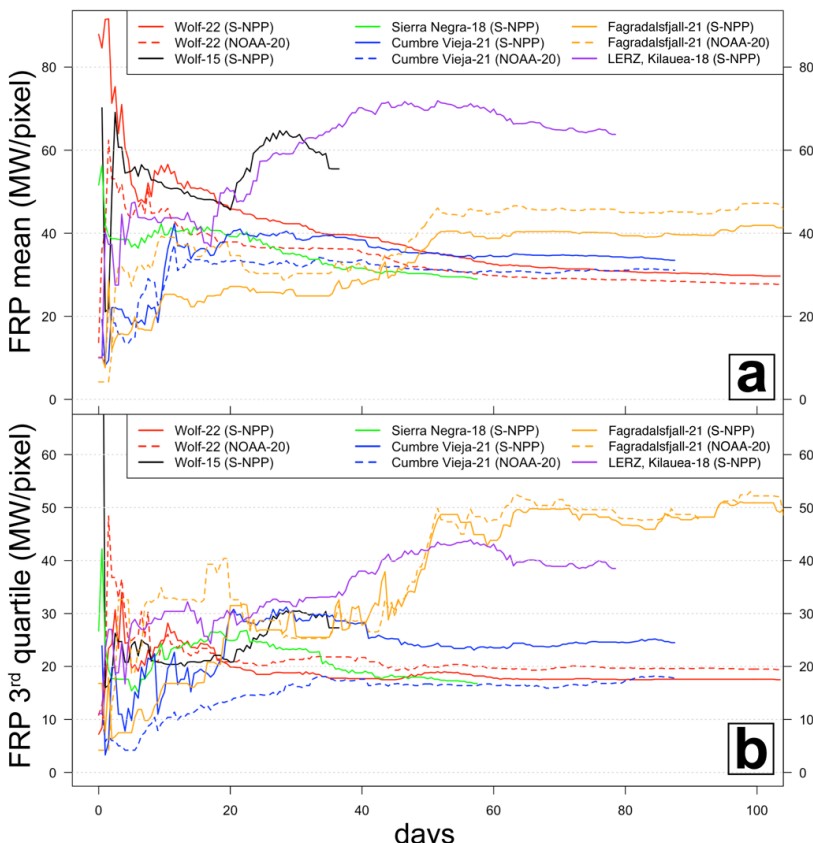

**Figure 4.** Cumulative (**a**) mean-FRP and (**b**) 3rd quartile-FRP over time for the six long-lasting eruptions being investigated, based on data retrieved every 12 h. Solid and dashed lines depict the data recorded by the S-NPP and NOAA-20 satellites, respectively. Note that NOAA-20 was launched at the end of 2019. This is why there is no data for earlier eruptions.

Importantly, statistical parameter variations were performed in a cumulative manner for threshold search since temporal variations of the non-cumulative FRP data every 12 h was found to fluctuate strongly throughout all long-lasting eruptions (Supplementary Material S3A), so no definitive value for a threshold can be obtained from the 12-h FRP analysis alone.

### 4.2. LavaFlow_Mapper

Based on the outcomes from the FRP statistical analysis, we generated four maps for each of the six long-lasting eruptions by using the 5th percentile, 3rd quartile, mean, and 95th percentile of the entire datasets as thresholds and compared them with their corresponding official map (white polygons in Figure 5).

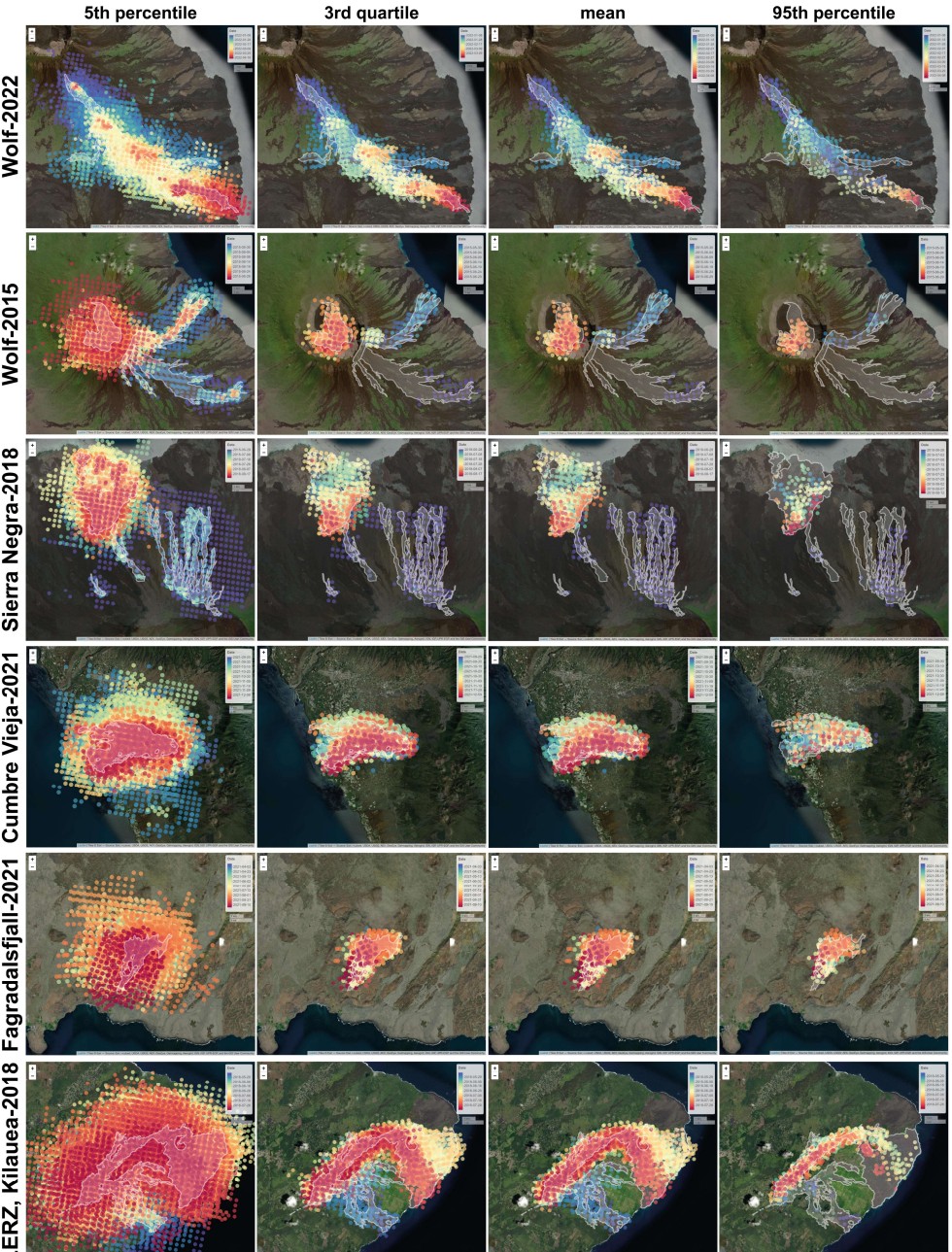

**Figure 5.** Composite figure of the six long-lasting eruptions being investigated. The white transparent polygon depicts the area covered by the lava based on the official lava flow maps. Colored dots highlight the date of acquisition of each thermal anomaly in the color spectrum scale, where violet/blue colors are the oldest, and the red ones are the youngest anomalies. Note that for Wolf-2022 a preliminary cut-off date of 20 April 2022 was chosen and that an official map has yet to be published (as of writing). To view a zoomed-in version of each case study, see Supplementary Material S2B.

As depicted in Figure 5, from left to right, the number of thermal anomalies shown on each map is reduced significantly, since the applied filters become more restrictive and an increasing number of lower FRP values is excluded. In all cases, when comparing the

resulting maps with the official ones, we observe that the 5th percentile map covers a significantly larger area than the official one. This is because the filter, on average, includes 48% of correct points (located within the official lava map) and 52% of noise (points outside the official map), i.e., ~1:1 ratio (Table 3). On the other hand, the 95th percentile filter produced maps that, in most cases, cover a much smaller extension than the official ones; since the filter value is so high, only a few thermal anomalies (<10%) surpass it (e.g., Kilauea in Figure 5 and Table 3). This filter includes, on average, 83% of correct points and 17% of noise (i.e., ~5:1 ratio). Consequently, higher value filters imply a better percentage of correct points, albeit at the cost of a significant reduction in the number of thermal anomalies, resulting in underestimated thermal maps (Figure 5). In between, the thermal maps resulting from using the 3rd quartile and the mean are much more similar to the official maps (Figure 5). For these thresholds, the correct points reach ~75%, and the signal-to-noise ratio is ~3:1. Moreover, the reduction of correct data points is significantly lower than when the 95th percentile filter is applied (Table 3). In general, the 3rd quartile is slightly lower (in ~15 MW) than the mean (Table 2) (i.e., more thermal anomalies are considered). Thus, the 3rd quartile filter could be considered as a threshold to produce slightly overestimated maps, while the mean produces marginally underestimated ones (Figure 5).

**Table 3.** Data reduction and error analyses after applying the FRP filters by counting the hot spots that are within and outside the official lava flow maps of the six long-lasting eruptions.

| **Data Reduction** | | | | | | | | | | |
|---|---|---|---|---|---|---|---|---|---|---|
| **Case Studies** | **No Filter (%)** | | **5th Per. (%)** | | **3rd Quar. (%)** | | **Mean (%)** | | **95th Per. (%)** | |
| | **Within** | **Outside** | **Within** | **Outside** | **Within** | **Outside** | **Within** | **Outside** | **Within** | **Outside** |
| Wolf-2022 | 7853 | 6032 | 7551 | 5283 | 2439 | 860 | 1855 | 606 | 602 | 163 |
| Wolf-2015 | 1220 | 1842 | 1187 | 1677 | 432 | 235 | 336 | 134 | 130 | 34 |
| Sierra Negra-2018 | 3059 | 2163 | 3024 | 1905 | 999 | 285 | 778 | 182 | 244 | 45 |
| Cumbre Vieja-2021 | 7655 | 8226 | 7511 | 7014 | 3168 | 725 | 2620 | 544 | 760 | 77 |
| Fagradalsfjall-2021 | 6367 | 7431 | 6184 | 6573 | 2269 | 612 | 2198 | 583 | 535 | 97 |
| LERZ, Kilauea-2018 | 7916 | 12349 | 7817 | 11451 | 3389 | 1513 | 2811 | 1056 | 951 | 225 |
| **Error Analysis** | | | | | | | | | | |
| **Case Studies** | **No Filter (%)** | | **5th Per. (%)** | | **3rd Quar. (%)** | | **Mean (%)** | | **95th Per. (%)** | |
| | **Correct** | **Noise** | **Correct** | **Noise** | **Correct** | **Noise** | **Correct** | **Noise** | **Correct** | **Noise** |
| Wolf-2022 | 56.6 | 43.4 | 58.8 | 41.2 | 73.9 | 26.1 | 75.4 | 24.6 | 78.7 | 21.3 |
| Wolf-2015 | 39.8 | 60.2 | 41.4 | 58.6 | 64.8 | 35.2 | 71.5 | 28.5 | 79.3 | 20.7 |
| Sierra Negra-2018 | 58.6 | 41.4 | 61.4 | 38.6 | 77.8 | 22.2 | 81.0 | 19.0 | 84.4 | 15.6 |
| Cumbre Vieja-2021 | 48.2 | 51.8 | 51.7 | 48.3 | 81.4 | 18.6 | 82.8 | 17.2 | 90.8 | 9.2 |
| Fagradalsfjall-2021 | 46.1 | 53.9 | 48.5 | 51.5 | 78.8 | 21.2 | 79.0 | 21.0 | 84.7 | 15.3 |
| LERZ, Kilauea-2018 | 39.1 | 60.9 | 40.6 | 59.4 | 69.1 | 30.9 | 72.7 | 27.3 | 80.9 | 19.1 |
| **Average** | **48.1** | **51.9** | **50.4** | **49.6** | **74.3** | **25.7** | **77.1** | **22.9** | **83.1** | **16.9** |
| **Std** | **7.5** | **7.5** | **7.9** | **7.9** | **5.8** | **5.8** | **4.2** | **4.2** | **4.1** | **4.1** |

Table 3 summarizes the results of the data reduction and error after applying various possible FRP filters. As mentioned before, the higher the filter value, the higher the relationship between correct points and noise, which is ~1:1 when no filter is applied and ~3:1 when the 3rd quartile and/or mean are used. At the same time, the number of thermal alerts that fall inside the official mapped areas are also reduced but are still roughly in the same order of magnitude of the original number of thermal alerts that were correctly located with no FRP filters. In contrast, when an even higher value, such as the 95th percentile is applied, the number of thermal alerts that is still being considered falls below ~10% of the original hot spots and is thus not representative anymore.

Moreover, we scrutinized the spatial error of the maximum linear extension of the lava flows obtained with the various filters and the official maps (Table 4). We found that when

no filters are applied, the overestimated relative error can vary between 0.3 and 84.4%, while when the filter of 95th percentile is used, it generates an underestimated relative error up to −20%. As before, when the 3rd quartile or the mean is used, the relative errors are significantly reduced to ±3% (Table 4).

**Table 4.** Relative error of the maximum linear extension of the lava flows based on comparison between the official maps and the results after applying the various FRP filters.

| Case Studies | No Filter (%) | 5th Per. (%) | 3rd Quar. (%) | Mean (%) | 95th Per. (%) |
|:---:|:---:|:---:|:---:|:---:|:---:|
| Wolf-2022 | 4.0 | 4.0 | −1.3 | −2.8 | −7.7 |
| Wolf-2015 | 0.3 | 0.3 | 0.2 | −5.5 | −7.5 |
| Sierra Negra-2018 | 5.7 | 5.7 | 2.0 | 0.4 | 0.0 |
| Cumbre Vieja-2021 | 39.3 | 36.1 | −1.3 | −1.3 | −4.8 |
| Fagradalsfjall-2021 | 84.4 | 84.4 | 0.7 | 0.7 | −24.3 |
| LERZ, Kilauea-2018 | 22.7 | 22.7 | 5.1 | 5.1 | −1.3 |
| **Average** | **26.1** | **25.5** | **0.9** | **−0.6** | **−7.6** |
| **Std** | **29.3** | **29.1** | **2.2** | **3.3** | **8.0** |

For the short-lived eruptions, most of the statistical parameters are higher than those of the long-lasting ones (Table 2). This implies that after applying the filters, a significant number of thermal anomalies of the already small datasets (between 26 and 528 thermal anomalies) would be excluded from the thermal map (Supplementary Material S2B). Therefore, better threshold alternatives for short-lived eruptions are discussed in Section 5.1.

## 5. Discussion

### 5.1. Short-Lived Eruptions: The Case of Fernandina Eruptions (Galápagos, Ecuador)

At the onset and during the first days of all analyzed eruptions, the FRP values vary strongly in both long-lasting and short-lived eruptions (Table 2 and Figure 4). This is due to the fact that at the initial stage of the eruption, discharge rates are generally very high, while they decay rapidly after a few days (Supplementary Material S1A). This decay in the cumulative FRP statistics curve can have a complex shape in its early stages, so that no stabilization is attained during the first days of ongoing and throughout short-lived eruptions. Therefore, since the statistical values rarely reached a stable level in short-lived eruptions and not enough satellite data acquisition was available for robust statistics and threshold determination, we explored some of the FRP thresholds obtained from the long-lasting eruptions to reproduce the area inundated by lava of the short-lived Fernandina eruptions (Figure 6).

Based on the relative accuracy of the 3rd quartile and the mean as thresholds previously observed in the long-lasting eruptions, we decided to use the averages of these two parameters (27 ± 10 and 39 ± 13, respectively, see Table 2) when running the LavaFlow_mapper code for the short-lived eruptions. Since the 3rd quartile and the mean vary in a narrow FRP range (between 17 and 52 MW) and they usually produce slightly underestimated or overestimated maps, we propose to use the midpoint of the two parameters (34.5, approximated to 35) and the resulting range (±17) to then produce the thermal maps of short-lived eruptions (Figure 6). Overall, when compared to the official maps, the areas covered with the thermal anomalies above the chosen thresholds fit the ones actually covered in lava. As mentioned before, lower values (17 MW) tend to overestimate the inundated areas while higher values (52 MW) underestimate them (Figure 6). Consequently, a filter of 35 ± 17 is a good approach for short-lived hotspot subaerial eruptions. It should be noted that the VIIRS spatial resolution is 370 m; consequently, our maps are always a rough approach to the area covered by fresh lava. This is especially true for short-lived eruptions that cover

small areas. Therefore, more extensive studies must be carried out in the future to assess the validity and usability of such a threshold in short-lived eruptions in similar environments.

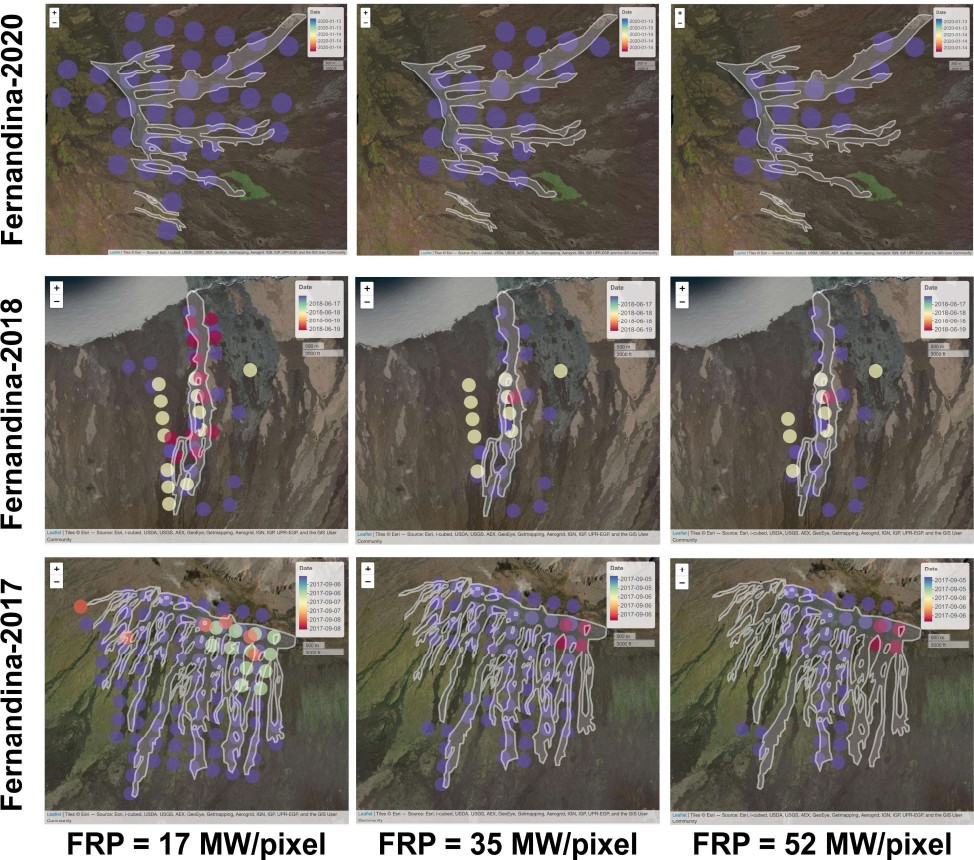

**Figure 6.** Composite figure of the three short-lived Fernandina eruptions by using the 35 ± 17 filter. The white transparent polygon depicts the actual area covered by the lava.

### 5.2. Ongoing Eruptions: Application to the Wolf-2022 Eruption (Galápagos, Ecuador)

In the previous sections, we discussed the difficulties of defining a suitable filter at the onset and during the first days of an eruption since the FRP data varies strongly in most of its statistical parameters (Figure 4 and Table 2). Therefore, as in the case of short-lived eruptions, for ongoing eruptions, we suggest the use of an FRP filter of 35, which is based on the midpoint of the global mean and 3rd quartile of the six long-lasting eruptions. We validated the proposed filter by applying it to the ongoing Wolf-2022 eruption, even though the 35 filter was obtained considering the data from this eruption. Since the Wolf-2022 eruption reached the statistical FRP cumulative stabilization after 30 days and its inclusion built a more robust dataset, we decided to include it when determining the filter. It should be noted that if Wolf-2022 data is disregarded from the average and mid-point analyses, the FRP filter increases to 37, implying a slight reduction of the filtered data (i.e., underestimation), but changes in the produced map are negligible.

Since there is no official lava map for this eruption, to do this, we took advantage of two clear high spatial resolution (3 m) satellite images from PlanetScope [15] of the Wolf-2022 eruption (Figure 7). Overall, Figure 7 shows that, compared to an image taken prior to the eruption, on both analyzed dates, i.e., at the beginning of the eruption (13 January, one week after the onset) and three months later (20 April), the area covered by the new lava (dotted line) is very similar to the area covered by the FIRMS thermal anomalies after applying the 35 FRP filter (yellow circles). However, there are some discrepancies at the lava fronts (underestimation) and at the margins (overestimation) (Figure 7, Tables 3 and 4). These differences could either be related to the size of the VIIRS pixels, which is 370 m compared to the 3 m of the satellite images, or they result from the weather conditions during the

satellite image acquisition (e.g., clouds). Another option could be related to the thickness of the lava flow and its associated thermal cooling process. For instance, in the case of Wolf-2015 eruption (Figure 5), Bernard et al. [27] reported that at the upper southeastern flank, the pahoehoe lava was most likely only a few centimeters thick, something that was also observed in Sierra Negra-2018 in the sheet flows of fissure two. The rapid cooling of such thin lava could significantly reduce the FRP values (lower than the 3rd quartile, mean and/or 35 MW) so that the areas covered by them would not be considered by our script (Figure 5; e.g., Wolf-2015: 5th per. vs. 3rd quar. and Sierra Negra-2018: 3rd quar. vs. mean).

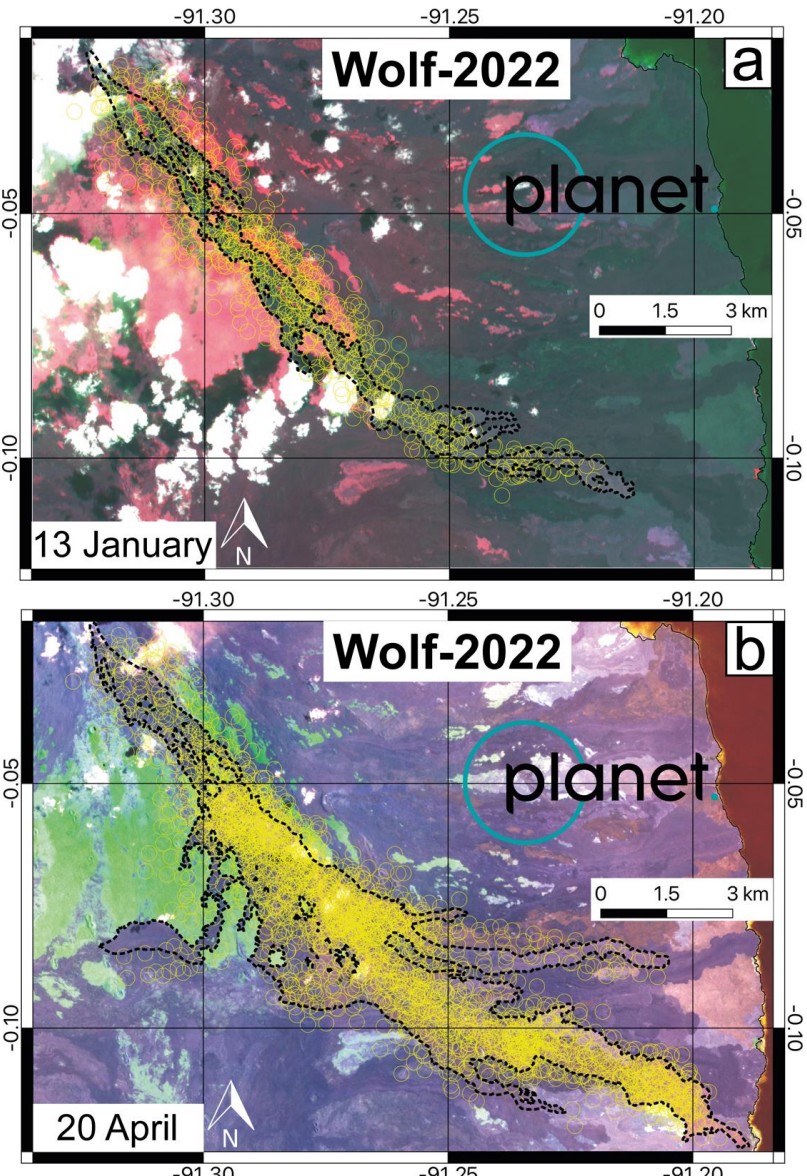

**Figure 7.** (**a**) PlanetScope false color composite image from 13 January 2022. (**b**) PlanetScope natural color composite image from 20 April 2022. The black dotted line depicts the area covered by the lava after comparing these satellite images to one taken prior to the eruption (18 May 2021). The yellow circles are the cumulative FIRMS alerts with FRP values above 35 until the same date.

### 5.3. Approximating Eruptive Breaks and/or End of Unobserved Eruptions

Defining the end of an eruption is a challenging task, especially for remote volcanoes, because of the lack of direct observations and the limited geophysical data. Some approaches have already been proposed for lava-forming eruptions that formed a Wadge curve by using the MODIS sensor [73]. The Wadge curve refers to eruptions that formed a

waxing/waning trend when plotting lava effusion rate against time [73]. At the onset of the eruption, there is a rapid increase in the lava flux (waxing flow), which then decays logarithmically over the rest of the eruption (waning flow) [74]. Along the same lines, we noted that, at some point, the incoming thermal alerts stop surpassing the FRP filter, and no new hot spots are included in the most current thermal map. Since values above the threshold were generally related to active lava and lower values to inactive ones, we explored the variations of the maximum FRP per day of the well-documented eruptions of Cumbre Vieja, Fagradalsfjall and LERZ-Kilauea—7 days before and after the verified end of the respective eruption (Table 1).

As depicted in Figure 8, the daily maximum FRP of these eruptions rapidly decays from hundreds to tens of MW/pixel (below the FRP filter), in less than three days after the official eruption ends. A similar decay is also observed at the end of the more remote long-lasting eruption of Wolf-2015. Additionally, a break in eruptive activity, where the maximum FRP fell below the chosen threshold for more than two consecutive days before rising again above the filter, was observed for the Wolf-2015 in the first days of June when the eruption changed from circumferential to intra-caldera (Supplementary Material S3B). This suggests that when the maximum FRP value per day rapidly decays in order of magnitude or more, reaching values below the FRP filter for a period longer than two days, a significant break in the eruptive activity or the end of the eruption can be inferred. Based on these considerations, we conclude that the unobserved Wolf-2022 eruption ended on 13 April, while the Sierra Negra-2018 most likely ended on 21 August, i.e., two days before the official report (Table 1). Moreover, since the uncertainty range is ±2 days, this method cannot be applied to infer the end of short-lived eruptions, such as those of Fernandina (Table 1). Finally, it should be noted that, as in the case of Fernandina-2017, thermal alerts can continue for a long time period after the actual cessation of an eruption when wildfires are sparked by lava [52]. Therefore, the eruption termination could go unnoticed when only thermal alerts are available for monitoring. Importantly, extreme weather conditions such as clouds, rain, and/or freezing temperatures could significantly reduce the maximum FRP values and, consequently, lead the observer to assume pauses or the end of unobserved eruptions mistakenly. Therefore, more monitoring parameters are needed.

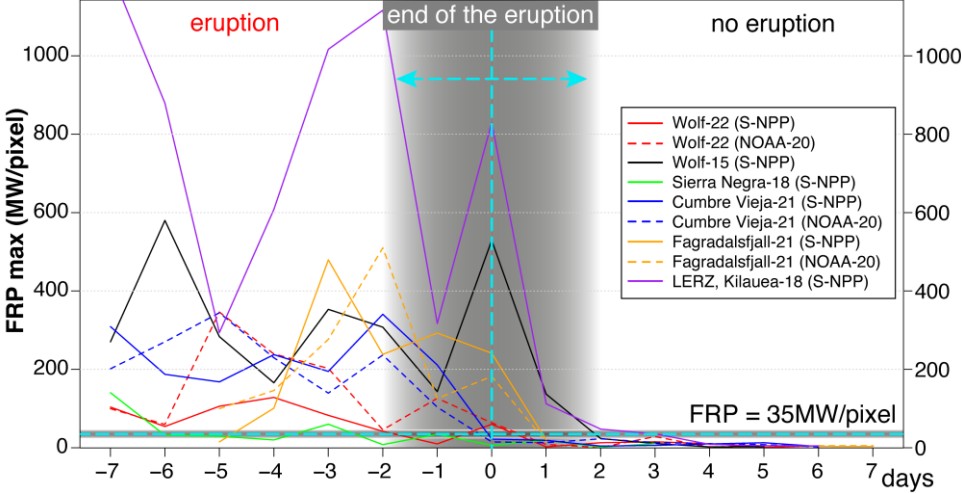

**Figure 8.** Analysis of the end of the eruption based on the max-FRP per day for the six long-lasting eruptions. The cyan dashed lines highlight the end date of the eruptions and the default FRP filter (35 MW/pixel). The shadowed area highlights the uncertainty range.

### 5.4. The Value of Time during Volcanic Crises

Time is key when volcano observatories respond to volcanic crises. In this context, we compared the computing times of our scripts on two desktop and two portable computers. We tried to cover the range of computers that could be found within a volcano observatory.

The basic characteristic of each one is detailed in Table 5. For calculating the computing time, we included the "tictoc" package in our scripts which was a timer that allowed measuring the time needed to read, analyze, process, plot, and save the information, and its result were shown in seconds.

**Table 5.** Basic characteristics of the computers used to test the computing time of our scripts.

| Characteristic | iMac (2017) | MacbookPro (2012) | PC (Desktop) | ThinkPad (2017) |
|---|---|---|---|---|
| Processor | 3 GHz Core-i10 | 2.5 GHz Core-i5 | 3 GHz Core-i7 | 1.6 GHz Intel Celeron |
| RAM memory | 32 GB | 16 GB | 8 GB | 4 GB |
| Storage | 1 TB | 256 GB | 500 GB | 116 GB |
| System | Monterey | Catalina | Windows 7 | Windows 10 |

Table 6 shows the results in seconds of the time it took to run each script on the different computers. Overall, computing time is in the range of less than 1 s to 5 min. The resulting computing times depend on the performance of each computer, but also on the amount of data that is analyzed. Thus, for long-lasting eruptions, with data of over thousands of observations, computing time lasted a few minutes, while for short-lived eruptions with data in the order of hundreds of observations, the scripts ran in less than two seconds (Table 6). Additionally, the FRP_statistical code took more time (maximum: 309 s or 5 min) than the LavaFlow_mapper (max. 7 s). This difference is mostly related to the 12 h cumulative statistical processing carried out by the statistical code. It should be pointed out that the FRP_statistical script does not need to be run every time a new thermal map is created, but rather just until a suitable FRP filter has been obtained. This is the case when stabilization of the cumulative statistical data is observed in the FRP_scatistical output. Until a robust database has been built, the default value of $35 \pm 17$ MW/pixel can be used directly in the LavaFlow_mapper code with a low risk of misrepresenting the eruption.

**Table 6.** Computing time after running the FRP_statistical and LavaFlow_mapper scripts on four different computers and for all nine case studies.

| Case Study | FRP Statistical (Seconds) | | | | LavaFlow_Mapper (Seconds) | | | | # Thermal Anomalies |
|---|---|---|---|---|---|---|---|---|---|
| | iMac | MacbookPro | PC | ThinkPad | iMac | MacbookPro | PC | ThinkPad | |
| Wolf-2022 [1] | 16.48 | 33.06 | 41.72 | 151.77 | 0.58 | 1.27 | 0.57 | 2.83 | 22,924 |
| Wolf-2015 | 2.27 | 5.86 | 5.29 | 18.72 | 0.31 | 0.31 | 0.19 | 0.97 | 5269 |
| Sierra Negra-2018 | 5.13 | 8.77 | 12.64 | 46.95 | 0.38 | 0.47 | 0.31 | 1.66 | 8628 |
| Cumbre Vieja-2021 | 25.8 | 45.28 | 65.6 | 238.56 | 0.83 | 1.39 | 0.76 | 5.45 | 30,285 |
| Fagradalsfjall-2021 | 32.35 | 57.28 | 85.25 | 309.43 | 0.65 | 1.08 | 0.67 | 4.85 | 19,576 |
| LERZ, Kilauea-2018 | 25.51 | 46.20 | 67.11 | 242.66 | 0.87 | 1.83 | 1.07 | 7.20 | 34,628 |
| **Average** | **18 ± 11** | **33 ± 19** | **46 ± 29** | **168 ± 106** | **0.6 ± 0.2** | **1.1 ± 0.5** | **0.6 ± 0.3** | **3.8 ± 2.2** | **$2.10^4 \pm 1.10^4$** |
| Fernandina-2020 | 0.59 | 0.22 | 0.19 | 0.72 | 0.19 | 0.12 | 0.10 | 0.30 | 73 |
| Fernandina-2018 | 0.16 | 0.26 | 0.22 | 0.98 | 0.20 | 0.12 | 0.11 | 0.31 | 392 |
| Fernandina-2017 | 0.26 | 0.27 | 0.24 | 1.51 | 0.20 | 0.14 | 0.11 | 0.39 | 528 |
| **Average** | **0.3 ± 0.2** | **0.3 ± 0** | **0.2 ± 0** | **1.1 ± 0.3** | **0.2 ± 0** | **1.1 ± 0** | **0.1 ± 0** | **0.3 ± 0** | **300 ± 200** |

[1] We took 20 April as preliminary cut-off date, while the exact date of the end of the eruption was still unknown. # stands for number.

Testing our scripts on different computers and operative systems proves that they can generate maps, figures, and associated tables in the range of a few seconds to minutes, which is especially relevant during crises. Even the slowest performing computer can generate maps in less than 10 s when processing more than 30,000 thermal anomalies.

### 5.5. Limitations, Advantages and Future Perspectives

The limitations of this study are specifically related to the spatial resolution, satellite conditions during image acquisition, and the amount of data. Regarding the first limitation, VIIRS pixel size is 370 m, which is significantly larger when compared with Landsat-8 (30 m), Sentinel-2 (10 m), or PlanetScope (3 m) satellite images. Therefore, our preliminary maps include a spatial error that sometimes is in the order of 1 km (Figure 5), i.e., rough mapping. Nevertheless, the S-NPP and NOAA-20 satellites pass twice per day and thus have a much higher temporal resolution than Landsat-8 (8 days), Sentinel-2 (once every 5 days), or PlanetScope (roughly once per day) satellites. The second limitation is related to the conditions during image acquisition, which include satellite layout and weather, which are both out of our control. In addition, since the mapper script applies filters on the layout and radiative power, when eruptions are short-lived, and the number of thermal anomalies is very low, few data overcome the filters, resulting in thermal maps with various gaps in areas actually inundated by lava (Figure 6 and Supplementary Material S2B). Importantly, since the FIRMS alerts detect hot anomalies, other heat sources, such as wildfires, can certainly alter the resulting thermal maps. Within our case studies, this was observed for the Fernadina-2017 eruption, in which the lava triggered wildfires that lasted over a month after the end of the eruption [52]. Beyond the limitations and given that our scripts were fed with the well-known, freely available, daily, and validated FIRMS datasets and that we took advantage of the open-source R software, which was especially useful for the statistical analysis and processing of big datasets, the advantages outweigh the limitations. Firstly, both the input data and the software to process them are fully free and easy to download and install (Supplementary Material S1). Secondly, the FRP statistical code allows going into detail on the statistical parameters of the fire radiative power (FRP), which is important to understand the raw data before choosing a filter. Moreover, the LavaFlow_mapper generates interactive maps with the additional benefits of (1) discriminating the advance of lava flows over time since the map includes a color-scale bar per acquisition date (Figures 5 and 6) compared to in-hindsight mapping, (2) having the possibility to change the scale (zoom in or out) on the basemap as needed, (3) changing the provider of the basemap for better visualization of the thermal anomalies (e.g., OpenTopoMap, OpenStreetMap and many others, see http://leaflet-extras.github.io/leaflet-providers/preview/index.html, last accessed on 13 June 2022), and the benefit that (4) the code takes only a few seconds to generate the map as well as the resulting figure and tables, even on low-performance computers (Table 6). Further, it should be pointed out that the filtering process and plotting can be done in other software such as: Excel, LibreOffice, ArcGIS, QGIS, and others. However, since our codes are especially designed to address the filtering and plotting processes, results can be obtained in a few seconds because most of the procedure has been automated. Based on our own experience, the alternative software usually crashes after managing that large amount of data, and if not, they at least take much longer. Additionally, further customization of statistical analyses, plotting parameters, and data presentation styles can be performed by more advanced R users if needed since the code is open.

Importantly, mapping lava flows is usually carried out by using satellite images, e.g., [28–30,52,61,75,76], overflights or unoccupied aerial vehicle campaigns (UAV), e.g., [55,56,58], all of which are often limited to a given budget and to weather conditions. This is the case of Wolf-2022 for which there are only four clear images in more than 100 days of the eruption. Regarding the latter, it appears that our codes are less affected by bad weather conditions, despite the fact that the FRP values provided by VIIRS sensors and their thermal band (0.412–11.5 μm) are lowered by the presence of clouds. Nevertheless, the use of relatively low thresholds on the LavaFlow_mapper code (e.g., 35 ± 17) when compared to the highest FRP values (e.g., average 95th percentile of 209 ± 122, Table 2) could prove useful in avoiding misestimation of thermal anomalies due to cloud coverage. Ideally, the use of our tool in combination with others, e.g., [28–30,76], will significantly improve the timely mapping of lava flows, which is essential for crisis management and

disaster risk reduction. Moreover, it should be pointed out that the suggested thresholds were tested on lava-forming eruptions of hotspot subaerial settings, and, consequently, for other eruptive styles and environments, future studies must recalculate suitable FRP filters.

Finally, it should be mentioned that, usually, volcanic thermal analyses are performed by using the MODIS sensor on board the AQUA and Terra satellites [9–11]. However, MODIS sensors are aging and nearing the end of their lives. Thus, most of the platforms that use MODIS started to compare the data with VIIRS sensors in order to be able to migrate from the former to the latter [77,78]. Moreover, volcanic studies generally use the volcanic radiative power (VRP), which is an adaptation of the FRP [77]. While here we use FRP values, these could easily be replaced by VRP ones in the future, if necessary, since our tool simply considers the raw numbers, and the only modification to our results would be regarding the threshold value (35 ± 17 MW) while maintaining the same procedure.

## 6. Conclusions

We took advantage of the near real time (updated every 12 h), validated and freely available FIRMS data and the open-source R software to develop a novel, simple, and free tool for rapid preliminary mapping of lava flows during ongoing hotspot subaerial eruptions. The tool is made up of two scripts, of which the first one statistically analyzes the FIRMS' fire radiative power (FRP) data to define a minimum threshold, which is then used in the second script. The mapper code uses the previously defined FRP and the track GSD thresholds as filters to plot the FIRMS thermal anomalies on top of a basemap to finally produce an interactive thermal map of the lava flows over time.

The first code, FRP_statistical, allowed us to characterize the FRP behavior throughout the nine eruptions being investigated. While at the onset and during the first days of an eruption, the FRP varies strongly; as the eruption continues, the cumulative statistical FRP parameters stabilize. For that reason, statistical FRP values of short-lived eruptions do not reach stability, while for long-lasting eruptions (>30 days), all cumulative parameters stabilize. Between the studied long-lasting eruptions, the minimum, 5th percentile, 1st quartile, and median are very similar, while the 95th percentile and maximum vary significantly. Importantly, this study also found that the mean and 3rd quartile, which vary in a narrow range between eruptions, are suitable FRP filters since their application implies the consideration of a moderate number of thermal anomalies.

The second code, LavaFlow_mapper, applies two filters, one on the FRP values (identified with the first code) and another on the track GSD (satellite layout), to produce dynamic thermal maps over time. By using the official lava flow maps of nine recent lava flow-forming eruptions and high spatial resolution satellite images, we found that a default filter of 35 MW/pixel was good enough to replicate the official maps of short-lived and ongoing eruptions. For long-lasting eruptions, a proper FRP filter can be determined by using the FRP_statistical code once the cumulative statistical FRP values stabilize. Moreover, we suggest using the mean or the 3rd quartile FRP values as filters, as our results have shown that their use allows producing similar thermal maps to the actual lava flow maps of the studied eruptions with an accuracy higher than 75% on average. Furthermore, the code can be indirectly used to infer a pause or the end of unobserved long-lasting eruptions when it is noticed that no new thermal alerts are included in the interactive map for more than two consecutive days.

Our codes have demonstrated that they can produce results in a short period of time (seconds), even on low-performance computers and for large amounts of data (from hundreds to tens of thousands of thermal anomalies), which is especially relevant during volcanic crises. The fact that these tools are all open access and open-source code means that researchers can modify and customize the various aspects easily with enough knowledge of the R statistical package.

Our tool constitutes an additional resource to generate preliminary maps of active lava flows over time. The tool does not have the intention to replace other validated, more sophisticated, and accurate methods (e.g., satellite images, UAV campaigns, overflights, etc.).

On the contrary, it is intended to provide a rapid and low-cost option, which is especially relevant in the context of effusive and remote volcanic crises, in particular for volcano observatories with financial limitations.

This is the first attempt to map active lava flows in hotspot subaerial eruptions by using the FIRMS datasets. For other lava-forming eruptions, other filter values must be scrutinized, considering other hot volcanic phenomena such as pyroclastic currents and ash plumes and environments.

**Supplementary Materials:** The following supporting information can be downloaded at: https://www.mdpi.com/article/10.3390/rs14143483/s1, Supplementary Material S1: Input data, software requirements, guide, and codes. Supplementary Material S2: FRP_statistical and LavaFlow_mapper results. Supplementary Material S3: Variation over time of the non-cumulative FRP statistics every 12 h and Maximum FRP every 24 h.

**Author Contributions:** Conceptualization, methodology, data curation, formal analysis and writing original draft, F.J.V.; data curation, formal analysis and writing original draft, J.C.A. and A.V.M.; conceptualization, validation and writing review and edition, B.B. and P.R. All authors have read and agreed to the published version of the manuscript.

**Funding:** This research received no external funding.

**Data Availability Statement:** Codes and examples are available in https://vhub.org/tools/lavaflowmapper/wiki, last accessed on 13 June 2022.

**Acknowledgments:** We acknowledge the use of data from NASA's Fire Information for Resource Management System (FIRMS) (https://earthdata.nasa.gov/firms, last accessed on 13 June 2022). We also thank the National Land Survey of Iceland, Institute of Earth Sciences—University of Iceland, Icelandic Institute of Natural History and the Icelandic Met Office for sharing the shapefiles of the area covered by lava flows during the Fagradalsfjall 2021 eruption, in particular Lovísa Mjöll Guðmundsdóttir. The first author also thanks Diego Narváez for introducing him to the R environment. We also thank the four anonymous reviewers an the editor for their thoughtful comments that improved the manuscript. This research was conducted in the context of IG-EPN's project "Generación de Capacidades para la Emisión de Alertas Tempranas" funded by Secretaría Nacional de Planificación y Desarrollo (SENPLADES).

**Conflicts of Interest:** The authors declare no conflict of interest.

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
