# Peer review of "A Near Real-Time and Free Tool for the Preliminary Mapping of Active Lava Flows during Volcanic Crises: The Case of Hotspot Subaerial Eruptions"

_remotesensing, doi:10.3390/rs14143483_

Round 1
Reviewer 1 Report
Clear and well written paper. Offers good options if you are not able to conduct on site observations.
Author Response
Reviewer #1:
Clear and well written paper. Offers good options if you are not able to conduct on site observations. Thank you for the comment, we really appreciate your assessment.
Reviewer 2 Report
The paper deals with an original near real-time tool to map active lava flows by using independent level-2 satellite-based products (Fire Radiative Power, from FIRMS dataset).
I found the paper original, interesting and well organized. I think the work could be of potential interest for a wide scientific community and certainly for Remote Sensing readers. I have a couple of major points and some minor issues I would like authors will address before publication.
1. Introduction
I think there is a recent bibliography authors should properly cite because it is in my opinin very relevant for this study. In particular, recently has been developed an open and free tool, based on Sentinel-2 and Landsat8/9 data, for mapping lava flows; it could be potentially used in combination with the proposed tool to check/validate the achieved results (also thanks to the higher spatial resolution of Sentinel and Landsat imagery). I strongly recommend authors to read (and properly cite) the following papers:
Francesco Marchese, Nicola Genzano, Marco Neri, Alfredo Falconieri, Giuseppe Mazzeo, Nicola Pergola (2019). A multi-channel algorithm for mapping volcanic thermal anomalies at a global scale by means of Sentinel-2 MSI and Landsat-8 OLI data. Remote Sensing, 2019, 11(23) 2876; https://doi.org/10.3390/rs11232876.
Genzano, N.; Pergola, N.; Marchese, F. A Google Earth Engine Tool to Investigate, Map and Monitor Volcanic Thermal Anomalies at Global Scale by Means of Mid-High Spatial Resolution Satellite Data. Remote Sens. 2020, 12, 3232, doi: https://doi.org/10.3390/rs12193232.
2. Materials and methods
Paragraphs 2.1 and 2.2 are mainly instructions to how to use the code. The scientific content here is very poor, thus I suggest authors to move these sections to the supplementary material or to an Annex.
Concerning the LavaFlow mapper (Paragraph 2.4) it is not clear to this reviewer if the vent location must be set or can be set, please clarify. Moreover, if not available, which is the "default value"?
5. Discussion
My first major point regards the FRP threshold set for short-lived eruptions (35 +- 17 MW) and its possible generalization to other eruptions and volcanoes. I think a more extensive study and investigation should be carried out to better asses the validity and usability of such a threshold in different eruptive styles and environments. Please, try to address this point or proper acknowledge this potential limitation.
My second major scientific issue is about the validation of this FRP-filter of 35MW (pages 15-16 and Figure 7). In fact, validation has to be carried out on a different and independent dataset, respect to the training one. But authors, to validate such a threshold, used the Wolf-2022 eruption which has been also employed to calculate the average FRP threshold! Moreover, the figure 7 is not very informative in that, as it is not clear how the lava flow area (dotted lines) are derived: it was a authors' personal interpretation? Or a specific test has been used to identify lava and discriminate from background? It is also not clear why Figure 7a is depicted in false color whereas Figure 7b is in natural colors. Please clarify this aspect which, in my opinion, is the actual scientific weakness of the paper.
Paragraph 5.4
I don't think this aspect being so crucial, as the FIRMS products are published and made available with some hours of delay respect to the sensing time. I think this paragraph could be significantly reduced.
Final considerations
The proposed tool is interesting and I think could have an utility, especially for those volcano observatories in remote areas and with limited economic possibilities. However, it should be described in a more realistic way, as it is based on a 370m pixel size and thus cannot produce detailed maps of lava flows. I suggest authors to describe it as a tool for "a rapid and rough mapping of lava flows", even in the title. Finally, it would be nice to have an acronym for the tool...please think about it.
Reviewer 3 Report
The writing is well organized and structured, but there are a few points to be revised and improved as below:
1) In Figures 3 and 7, it needs north or south orientation if applicable;
2) It is not clear for a similarity percent report or information. If possible, it is better to provide a similarity report.
In summary, it needs a revision.
Author Response
Reviewer #3:
The writing is well organized and structured, but there are a few points to be revised and improved as below:
1) In Figures 3 and 7, it needs north or south orientation if applicable. Done
2) It is not clear for a similarity percent report or information. If possible, it is better to provide a similarity report. Thank you for the comment but we don't really understand what do you mean for similarity reports. If you are referring to validation, we understand the importance of assessing the tool with other independent methods. That is why we included a validation report in section 4.2 where we compared the results of using our LavaFlow_mapper with the lava maps published by the official institutions at each country at the end of the eruption (see also tables 3 and 4).
Reviewer 4 Report
Dear Authors,
I do really appreciate your paper and I do not have significant comment or request to move.
I added very few comments in the attached file that can slightly improve its content.
Thanks

Author Response
Reviewer #4:
I do really appreciate your paper and I do not have significant comment or request to move. I added very few comments in the attached file that can slightly improve its content.
Line 71: The VIIRS instrument provides 22 spectral bands at two spatial resolutions, 375 meters (m) and 750 m, which are resampled to 500 m, 1 km, and 0.05 degrees in the NASA produced data products. Could you add which bands have you used and could also better specify which GSD have you used? Done. We included the band used by FIRMS in the Materials and Methods section, since we are using a level-2 satellite-based product, i.e., we are not directly working with the band datasets. It reads as follows: “The Visible Infrared Imaging Radiometer Suite (VIIRS) instrument on board the joint NASA/NOAA satellites Suomi National Polar-orbiting Partnership (S-NPP), with data since 2012, and the NOAA-20/JPSS-1, with temporal coverage since 2020 [28,29], provides continued Fire Radiative Power (FRP) data twice per day with a pixel size of 370 meters [7,30,31] by using the thermal band 0.412 - 11.5 um.” Moreover, the GSD that was used was the track GSD following the recommendations of Wang et al., 2017. Therefore, we included the use of the term GSD (ground sample distances) after track in section 2 and when possible throughout the manuscript.
Line 102: It is not clear to me how do you define the reliability of your results. It is an authomatic process? Is it a comparison with external data/map? is it based on the stastical approach described in the follow?. We measure reliability by comparing the results of our tool with the official lava flow maps (see section 4.2). However, during new case studies, especially for ongoing eruptions where there are no maps, satellite imagery (optical or radar) must be used to defined reliability (see section 5.2), which must be done by the user. It is important to note that our scripts are a complementary tool for the rapid and preliminary mapping of active lava flows, and we do not have the intention to replace other more accurate, sophisticated, and validated methods to map lava flows. We add this sentence for clarification in the caption of figure 2 “In this context, reliable is defined by comparing the results of our tool with satellite or UAV images.”
Line 329: I suggest to visit the following links
http://www.labs.cs.ingv.it/index.php/view/map/?repository=kilauea&project=Kilauea
this webGIS include the results of the real time monitoring performed during the 2018 event and reported in a paper published in 2021
Musacchio, M., Silvestri, M., Rabuffi, F., Buongiorno, M. F., & Falcone, S. (2021). Kīlauea–Leilani 2018 lava flow delineation using Sentinel2 and Landsat8 images. Geological Society, London, Special Publications, 519.
And http://www.cos.cs.ingv.it/index.php/view/map/?repository=lapalma&project=lapalma
Thank you for the comment. We included the reference in the introduction and discussion sections. We also added the following sentence in the 5.5 section. “Ideally, the use of our tool in combination with others, e.g., Musacchio et al., 2021; Poland et al., 2022; Genzano et al., 2020; Marchese et al., 2020, will significantly improve the timely mapping of lava flows, which is essential for crisis management and disaster risk reduction.”